# A Macaque Brain Extraction Model Based on U-Net Combined with Residual Structure

**DOI:** 10.3390/brainsci12020260

**Published:** 2022-02-12

**Authors:** Qianshan Wang, Hong Fei, Saddam Naji Abdu Nasher, Xiaoluan Xia, Haifang Li

**Affiliations:** 1College of Information and Computer, Taiyuan University of Technology, Yingze Street, Taiyuan 030024, China; wangqianshan0203@gmail.com (Q.W.); feihong0419@link.tyut.edu.cn (H.F.); sadamnagi5@gmail.com (S.N.A.N.); 2Centre for Cognitive and Brain Sciences, University of Macau, Avenida da Universidade, Taipa, Macau SAR, China; xlxia@um.edu.mo

**Keywords:** brain extraction tool, macaque brain MRI, data fusion, residual structure, U-Net application

## Abstract

Accurately extracting brain tissue is a critical and primary step in brain neuroimaging research. Due to the differences in brain size and structure between humans and nonhuman primates, the performance of the existing tools for brain tissue extraction, working on macaque brain MRI, is constrained. A new transfer learning training strategy was utilized to address the limitations, such as insufficient training data and unsatisfactory model generalization ability, when deep neural networks processing the limited samples of macaque magnetic resonance imaging(MRI). First, the project combines two human brain MRI data modes to pre-train the neural network, in order to achieve faster training and more accurate brain extraction. Then, a residual network structure in the U-Net model was added, in order to propose a ResTLU-Net model that aims to improve the generalization ability of multiple research sites data. The results demonstrated that the ResTLU-Net, combined with the proposed transfer learning strategy, achieved comparable accuracy for the macaque brain MRI extraction tasks on different macaque brain MRI volumes that were produced by various medical centers. The mean Dice of the ResTLU-Net was 95.81% (no need for denoise and recorrect), and the method required only approximately 30–60 s for one extraction task on an NVIDIA 1660S GPU.

## 1. Introduction

Due to the restrictions on direct research of the living human brain, the research team took a different approach, in order to derive the working principles of the human brain, through the study of various animal brains [1,2]. Rhesus macaque (Macaca mulatta) is a typical nonhuman primate. Their brain structure is similar to humans and can be used as a transition model of the human brain [3]. Using the macaque brain to explore the working and pathological mechanisms of the human brain has become an essential method in human brain research [2,4,5,6,7].

Brain tissue extraction is a fundamental step in studying brain structure and function through brain medical images [8,9,10,11]. Brain tissue extraction focuses on removing non-brain tissues, such as the skull, muscles, and eyes, and then preserving the brain tissue [12,13,14]. Numerous software tools, for use on the human brain for skull stripping, have been developed, such as the Brain Extraction Tool (BET) in FSL [15,16], 3dSkullStrip in AFNI [17,18,19], and the hybrid watershed algorithm (HWA) in FreeSurfer [20,21]. Although these tools can perform well, when applied to the human brain, their performance is lacking when used on the macaque brain, mainly due to the image differences between the macaque and human brains [22]. As shown in Figure 1, in addition to the significant differences in size and shape, macaques also show the eyes as more prominent than those of humans in MRI images; there is more fatty tissue and a thicker skull around the brain, and the frontal lobe of macaques is very narrow and prominent [23]. Therefore, although both HWA and AFNI have optimized settings for nonhuman primates, these methods still require many manual settings. The inspections and corrections of the results are also time- and labor-consuming.

At the same time, deep neural networks have shown excellent performance in various computer vision tasks. It has been widely used in medical image segmentation [24,25,26]. An increasing number of studies have proven the performance of CNNs in the extraction of human brain tissue [27,28]. It inspires us that using deep learning may perform well in macaque brain extraction. CNN loses image details in convolution and pooling. Because the feature map size becomes gradually smaller, it cannot point out the specific contour of each pixel well. For this problem, a full convolutional network (FCN) structure was proposed [29]. Then, 3D FCN methods, to completely use the spatial structure information of medical images for segmenting human brain structure, were proposed in 2018 [30,31]. Meanwhile, U-Net has achieved impressive results in medical image segmentation. U-Net and FCN have a similar structure, but not the same. Compared with FCN, U-Net has a completely symmetrical encoding and decoding structure and concatenation skip connection. U-Net cascades the shallow features for accurate segmentation and depth features, for location, through encoders and decoders, which can effectively solve the problems of blurred boundaries, complex gradients, and the lack of high-resolution information in medical images segmentation. It realizes the feature operation of a larger dimension, through feature stitching, and adapts well to the training of limited sample data [32,33]. Deep learning can achieve excellent human brain tissue extraction performance, benefiting from the many high-quality human brain MRI images currently available [34]. However, as shown in Figure 2, macaque MRIs are usually obtained with various magnetic fields (1.5 T, 3 T, 4.7 T, 7 T, and 9.4 T), and the captured brain tissue signals are significantly different. Under high-fields MRI, nonstandard surface coils were also used, which were accompanied by deformation problems [35]. Furthermore, different sites use different types of equipment and protocols for acquisition. That leads to the suitable macaque brain MRI image data being rare, with significant differences in the quality and characteristics of the data between and within sites. Accordingly, except for the information detail loss, the problems of multiple centers and limited samples (on the macaque data) also required attention.

Transfer learning is a training strategy that speeds up model training, reduces data requirements, and prevents overfitting [36]. This strategy is used to train the extraction of human brain white matter signals, so that the model can be well-adapted to human brain MRI data under different acquisition protocols [37]. Humans and macaque are both primates, and the MRI signals from their brain tissue are similar. Using transfer learning, the model training for the macaque brain extraction method can achieve better performance [38]. However, due to the multicenter and limited sample characteristics of the macaque brain tissue data, mentioned above, the target domain’s feature distribution is unbalanced in applying transfer learning. These problems cause the models to have poor generalization performance and limit the extraction of subject sites not involved in the training dataset.

This research focuses on better using the structural similarity between humans and macaque and training deep learning networks, under multicenter and limited sample data, in order to obtain models with better generalization performance. At the same time, it is necessary to prevent overfitting, which occurs very often when the model is trained on limited sample data. Choosing an appropriate data training strategy and feature retention structure has become the key to solving the above problems. This study aims to make contribute to the professional field by applying the following methodologies. First, using T1w and T2w human brain MRI images to train the pretraining model. It makes the model pays more attention to the spatial correlation characteristics of the brain structure, rather than the signal peak distribution. Then, the weight of the human brain MRI extraction model, as the pretraining model, is applied to macaque brain MRI, through transfer learning, in order to improve macaque brain extraction accuracy. Significantly, a residual structure in U-Net was added, in order to obtain a ResTLU-Net model, retain the information lost between different layers, enhance the model training effect, and reduce overfitting. It further improved brain tissue extraction accuracy and enhance the model’s generalization ability on the limited dataset.

## 2. Materials and Methods

### 2.1. Dataset

The human brain MRI dataset (Dataset I) was obtained from the Human Connectome Project (HCP) [39], WU-Minn 1200 subjects data release. The preprocessing pipeline of the dataset mainly uses the FSL and FreeSurfer tools. This research uses the brain T1w and T2w image pairs of 67 subjects from the dataset, as well as their corresponding brain tissue mask files, as the extraction ground truth to create training and validation datasets.

The macaque dataset (Dataset II) was obtained from PRIME-DE (PRIMatE Data Exchange consortium) [40]. PRIME-DE contains 25 datasets from 22 sites. Detailed information about the datasets, provided by PRIME-DE, is found on the website (https://fcon_1000.projects.nitrc.org/indi/indiPRIME.html, accessed on 20 January 2022). This research uses the T1w images of 20 macaques from 10 sites from the dataset and manually labels their brain tissues as the extraction ground truth to create training and validation datasets.

### 2.2. Resampling

The macaque dataset used in this research comes from PRIMatE. The dimensions of the data, collected by each site, vary significantly (for example, 480×512×512, 200×256×256). To reduce the computational cost and obtain a consistent input image, while ensuring that the network can learn adequate information, the research team set the resampling layers before the input and output of the network. At the beginning of training, each slice is resampled to a size of 256×256. The intensity values of the MRI slices used in the experiment are substantially different, so normalization makes the average intensity value and variance close to 0 and 1. The image will be restored to its original size. Finally, the MRI data used in the experiments were not registered to standard space to further verify model’s generalization performance.

Figure 3 illustrates the training process, proposed in this research, to improve the extraction of macaque brain tissue by using a transfer learning strategy. The process consists of two steps. The first step is the construction of the pretraining model. This step primarily uses the model to benchmark and use the paired human brain T1w and T2w images for training, in order to generate a pretraining model. It may make the parameters in the pretraining model pay more attention to the spatial structure characteristics of the brain tissue image, rather than the signal peak distribution. The next step is constructing the macaque brain extraction model using transfer learning. This part trains and evaluates the model, using T1 images of macaques for validation. The specific training process is as follows:Dataset I is trained for 10 epochs to generate a pretraining model. After that, the best performing pretraining model is selected through the validation dataset for subsequent training of macaque images.The model will be trained on Dataset II for 40 epochs, and the best-performing model from the validation set is selected as the final model for testing. Because there were apparent differences between the MRI datasets collected for macaques and humans, the weights of all layers were updated on the pre-trained model.

This article will apply the training strategy to all models that need to be used for transfer learning.

### 2.3. U-Net Structure

The structure of the U-Net model is shown in Figure 4. The left part of the figure is the convolutional encoder, and the right part is the convolutional decoder. The convolutional coding structure is used to extract information from the input MRI image, capture the information between contexts, and encode it to detect edge information and features in the MRI image. The encoder is used for the precise positioning and reconstruction of the details. Both the encoder and decoder use basic convolution operations. There are five blocks in the encoder, and each block is composed of two convolutions. After each convolution, the layers are connected with normalization and the ReLU activation function. After each block completes its operation, 2×2 maximum pooling is used to downsample the image. During downsampling, the number of channels grows from 3 to 256, and the image dimensions shrinks from 256×256 to 16×16. The decoder process contains four blocks. Each decoder block contains a permutation convolutional layer and ReLU activation function. The number of channels is reduced from 256 to 2, and the image size is restored from 16×16 to 256×256. In the decoder process, to compensate for the loss of information caused by the pooling operation, each layer of the transposed convolution operation is connected with the feature map of the corresponding size from the encoding process. Finally, the research team utilized a 1×1 convolutional layer to map the final feature map into a two-class feature map.

The model uses a binary cross-entropy loss function, making the training more stable than the Dice loss function, and Equation 1 expresses the loss. The Adam algorithm is used to optimize the gradient descent algorithm (learning rate initialized at 0.0001, exponential decay rate of first-order moment estimation = 0.9, and exponential decay rate of second-order moment estimation = 0.999):(1)$$L=\frac{1}{N}\underset{i}{\sum }-\left[y_{i}log\left(p_{i}\right)+\left(1-y_{i}\right)log\left(1-p_{i}\right)\right]$$
where $y_{i}$ is the label of sample i, the positive type is 1, the negative type is 0, and $p_{i}$ is the probability that the sample is predicted to be in a positive class.

Although the U-Net model can solve the problem of model training difficulties, caused by limited sample data, when the data has multiple centers and limit samples simultaneously, the generalization ability of the model with different data is insufficient. This is mainly because the deep learning network loses a large amount of low-dimensional spatial information in the process of high-dimensional feature extraction. However, low-dimensional spatial feature information in medical images may play a more critical role in medical image segmentation. Therefore, retaining low-dimensional spatial feature information has become the focus of improvement.

### 2.4. ResTLU-Net

In deep neural networks, the model can theoretically obtain better results as the depth increases. However, as the network deepens, the gradient vanishes and other issues make continued training challenging [41]. The residual unit is the core of ResNet. With a block as the unit, each block comprises a series of layers and a shortcut. The shortcut crosses the middle layer, connects the input and output, and performs a simple equivalent mapping [42]. It does not produce redundant network parameters and calculations, enabling deepened network performance. When the input is $x$, Equation (2) is used to express the residual structure:(2)$$x_{i+1}=f_{i}\left(x_{i}\right)+x_{i}$$
where $f_{i}\left(x_{i}\right)$ is a nonlinear transformation function, combined with a convolution operation, batch normalization, and ReLU. Equation (3) expresses the general form of $f_{i}\left(x_{i}\right)$ in a convolution network:(3)$$f_{i}\left(x_{i}\right)=W_{i}\cdot \sigma \left(B\left(W_{i}^{\prime }\cdot \sigma \left(B\left(x\right)\right)\right)\right)$$
where $W_{i}$ represents the weight matrix, · represents the convolution operation, B(x) is the batch normalization, and σ(x) is the nonlinear activation function.

The residual network can integrate different subnetworks, and there is no dependency between the subnetworks [43]. Furthermore, these subnetworks consist of multiple shallow networks. Therefore, the residual structure can effectively retain low-dimensional spatial information and combine it with high-dimensional abstract features, fully enriching the computational information in the network and improving model training. Thereby, the loss of information in training is reduced, training effect is improved, and model generalization ability is enhanced. Since shallow information is crucial for segmentation, the research team replaced all first layer convolutions of each block in Figure 4 with a residual structure, in order to form a ResTLU-Net model. The ResTLU-Net model structure is shown in Figure 5.

For the U-Net model, the input of each block is $x_{i}$, and the output $y_{i}$ of each block can be expressed via Equation (4).
(4)$$y_{i}=f_{i+1}\left(f_{i}\left(x_{i}\right)\right)$$

Then, the output $y_{i}$ of each ResTLU-Net block can be expressed by Equation (5). It is expressed by how much low-level information is retained and combined into higher-dimensional information.
(5)$$y_{i}=f_{i+2}\left[f_{i+1}\left(f_{i}\left(x_{i}\right)+x_{i}\right)\right]$$

### 2.5. Experiment and Evaluation

The experiment in this research used the Ubuntu 18.04 operating system and a GeForce RTX 1660S GPU. The networks were coded with Python and the PyTorch framework. The experiment focused on the generalization ability of the model for multicenter dataset, so 8 of 10 sites macaque images (16 subjects) were randomly selected as the training and validation sets during the training process. The remaining two sites (four subjects) macaque images were used as the test set. The results of the pixel classification are clearly shown in Figure 6.

This research used quantitative indicators, commonly used in image segmentation, to evaluate the performance of brain tissue extraction methods. The indexes used to compare the extraction performance were Dice, Sensitivity (TPR), Specificity (TNR), Jaccard, voxel overlap error (VOE), false-negative rate (FNR), false-positive rate (FPR), and precision positive voxel (PPV). The equations for eight of the evaluation are as follows.
$$\begin{array}{c} Dice=\frac{2TP}{2TP+FP+FN}\;TPR=\frac{TP}{TP+FN}\;TNR=\frac{TN}{TN+FP}\;Jaccard=\frac{TP}{FN+TP+FP} \end{array}$$
$$\begin{array}{c} VOE=1-\frac{TP}{FN+TP+FP}\;FNR=\frac{FN}{FN+TP+FP}\;FPR=\frac{FP}{FN+TP+FP}\;PPV=\frac{TP}{TP+FP} \end{array}$$

In order to evaluate the performance of the model used in this research on dataset II and avoid over-fitting, caused by the selection of the training dataset, the five-fold cross-validation method was used to test the model. The data split process for the five cross-validations is shown in Figure 7. The mean Dice of the five-fold cross-validation results represents the final performance of the model.

## 3. Results

### 3.1. Comparsion of U-Net and ResTLU-Net

The pretraining model was trained on human brain images. Figure 8 shows the loss value, obtained by using the U-Net and ResTLU-Net models on Dataset I. After training for 10 epochs, the loss function converged. Therefore, only 10 epochs were trained, in order to avoid overfitting and choose the best model as the pretraining model.

Figure 9a shows U-Net’s training loss function value change graph (on different training strategies) and ResTLU-Net. Compared with the U-Net network without our transfer learning, the U-Net network with transfer learning has much faster loss function convergence. Furthermore, ResTLU-Net with the residual structure had a more stable loss during the training process. Compared with U-Net, it converged earlier, and the final loss value was smaller than the U-Net model. Figure 9b shows the Dice coefficients of each training epoch of different networks, valued on the validation dataset. The Dice of the models using our transfer learning strategy show much higher coefficients, and the performance tended to be more stable, which means our transfer learning strategy could improve the network performance in the extraction of macaque brain tissue. Meanwhile, there was severe overfitting in the U-Net network (from the change of validation accuracy). However, the Dice of ResTLU-Net fluctuated less on the validation dataset, and the overall result was stable. Moreover, the Dice of ResTLU-Net was higher than U-Net combined with our transfer learning strategy.

Finally, 10 macaque images, from different sites not involved in the training, were used as the test set to evaluate DenseNet, LinkNet, U-Net, TLU-Net, and ResTLU-Net. The results are shown in Table 1. Using U-Net as the basic network, the trained model was significantly better than DenseNet and LinkNet in brain tissue extraction. Therefore, further improvement of the U-Net network structure becomes the preferred solution.

The Dice of ResTLU-Net was 2.98% higher than that of U-Net and 1.68% higher than that of U-Net with transfer learning. In addition, ResTLU-Net is improved in TPR, Jaccard, VOE, and FNR. While the FNR was reduced, there was no significant change in FPR, indicating that the part that was not correctly extracted into brain tissue was correctly extracted, and no severe over-segmentation problems occurred.

### 3.2. Verification of the Validity

For this research, a five-fold cross-validation for the mentioned models was performed, and the results of the validation are shown in Figure 10. In the five training sessions, the decreasing trend of the loss function of the U-Net model (with transfer learning) at the beginning of training was much greater than that of the U-Net model. Compared with the U-Net model (with transfer learning), the ResTLU-Net model had minor loss values during the training process, and its downward trend tended to stabilize.

Table 2 shows the results obtained by testing three different models on five sets of the same cross-validation dataset. Compared with U-Net (with transfer learning), the mean Dice of ResTLU-Net increased by 1.11%; compared with U-Net, it increased by 4.12%. The results show that ResTLU-Net can maintain a relatively stable extraction effect for different training data, and both are higher than U-Net (with or without transfer learning). The average accuracy rate varied within 0.04%. This shows that ResTLU-Net has better generalization performance in the extraction of macaque brain tissue.

### 3.3. Comparison of ResTLU-Net and Other Brain Tissue Extraction Methods

This research compares the proposed model’s macaque brain tissue extraction results with other brain tissue extraction tools. Figure 11 show the segmentation results of the same batch of macaque brains by different brain tissue extraction methods. The neural network model has a higher and more stable performance than other traditional methods. 

As shown in Figure 12, the traditional U-Net does not perform well on 4.7 T macaque brain MRI data, and there is vastly erroneous segmentation. The addition of a transfer learning strategy can effectively improve the correct segmentation rate of the model, but there are still apparent errors and sizeable missing areas in the data of 3 or 4.7 T. The ResTLU-Net with residual structure overcomes these problems. These results show the feasibility of using limited samples for transfer learning, as well as the effectiveness of adding a residual structure to the network.

## 4. Discussion

This research reveals that using transfer learning strategy and residual structure to realize medical image recognition and segmentation of limited sample data is compelling. Transfer learning can provide a better pre-training model for faster and stable training progress in limited sample data. Adding the residual structure of short-hop links can effectively improve the feature extraction ability of the network, which is very suitable for the characteristics of multicenter and limited sample macaque data. Finally, the proposed method can achieve the apparent advantages of simple operation and fast processing speed, compared with other traditional tools, saving the data processing time for relevant researchers.

The current research has some limitations. First, only human and macaque MRI data were used. In future research, other non-human primate species, such as chimpanzees and long-tailed monkeys, will be considered to improve the generalization ability of neural network models and, finally, produce a more universal primate brain recognition model. Secondly, the MRI data concentrated on 1.5, 3, and 4.7 T in this research. Due to the lack of ‘gold standards’, high-field MRI data, such as 7, 9.4, 11.2 T, have not been carried out in enough experiments. However, according to the research team’s experience in processing medical image data, the signal sensitivity of different brain tissues will be different under different field strengths, and the signal characteristic distribution of the captured MRI data will be different. Therefore, whether the model proposed in this study can be effectively implemented in high-field MRI data remains to be further verified. However, adding a relevant data training model may help the model improve its generalization ability. At the same time, there is a lack of T2 weighted images and their corresponding segmentation ‘gold standards’ in the macaque brain tissue data collected by the research team. This study did not add pairs of T1- and T2-weighted images for the training process of the macaque brain. Although the current model has achieved relatively satisfactory performance, the results in the pre-training stage showed that it will be helpful to improve the accuracy and ability of model generalization by adding T2-weighted images as a supplement in the training stage. 

The above limitations belong to the research problem of improving the quality of the datasets. When the 112 macaque brain MRI, shared by Xindi Wang, is used as a dataset for training, the stability and generalization ability of the model was further improved. The average tested on independent data can be stable at 0.9739 ± 0.0084, which further proved that our ResTLU-Net design is reasonable. In the next stage, the research team will study the primate MRI data collection and processing standards, in order to improve the data quality of datasets. In addition to directly improving the tissue extraction method, adding the suitable method to optimize the data preprocessing stage, automatically identify the types of noise interference in the data [25], and selecting the appropriate processing method thereby improves the data quality. 

At the same time, the current 3D patches and 2D slices used in medical image segmentation networks lack critical spatial information [31]. This research used 2D slicing to achieve fast and reliable extraction, with minimal computing power. Feeding high-resolution, whole-brain MRI images directly into 3D CNNs can theoretically introduce more information to achieve better performance [30,44], but these methods are currently tricky. Some researchers have proposed methods to solve the memory problem of whole-brain segmentation, such as using different CNNs to work with different overlapping brain regions. However, most of these methods require complex data preprocessing steps, easily leading to errors. Due to the complex 3D architecture and spatial dependence between slices, 3D CNNs have significant advantages in 3D medical images. Reducing the calculation load of 3D CNNs may be the next research hotspot in medical image segmentation.

Furthermore, pre-training plays a vital role in model convergence and generalization, so pre-training models can also be used for multimodal image segmentation. Further research on pre-training models for brain tissue extraction or segmentation across multiple species is also possible.

## 5. Conclusions

Although the traditional U-Net model has been successfully applied in medical image segmentation, it is rarely applied to multicenter and sample-limited macaque data. Compared with other work that used U-Net as the fundamental network framework, the research team used different transfer learning strategies and network improvement methods to propose an improved ResTLU-Net model with its training process. (A) The human brain pre-training model uses paired T1 and T2 weighted MRI data. It makes the pre-training model pay more attention to the spatial structure of the tissue. (B) Optimize the network using model weights, trained on paired human brain MRI data models. (C) The residual structure is added to U-Net to help the model obtain high-dimension abstract feature information and low-dimension spatial feature information simultaneously during training. ResTLU-Net model performs image segmentation tasks can reach stable Dice coefficients to 0.9509 ± 0.0356, with no need for denoising, recorrecting, and registration. Finally, ResTLU-Net model overcomes the shortcomings of traditional segmentation tools that are time-consuming and labor-intensive. ResTLU-Net only takes 30–60 s to extract complete brain tissue on NVIDIA 1660S GPU. Therefore, the proposed model can complete segmentation quickly and effectively, while avoiding complex pre- and post-processing procedures, which is significant to brain science research.

## Figures and Tables

**Figure 1 brainsci-12-00260-f001:**
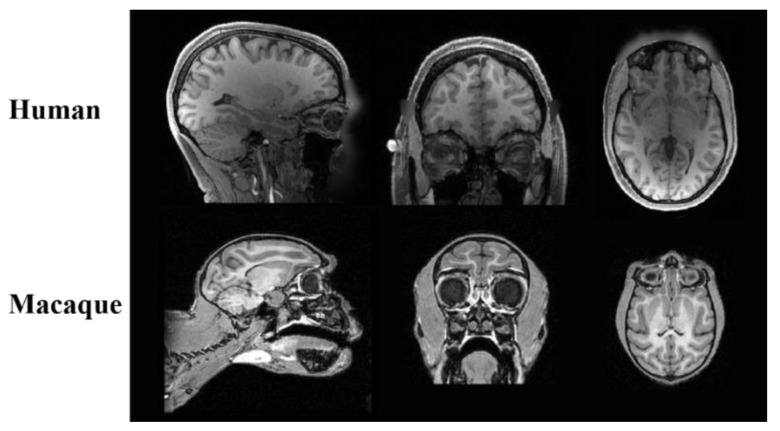
Schematic diagram of tissue structure difference between human and macaque brain in MRI.

**Figure 2 brainsci-12-00260-f002:**
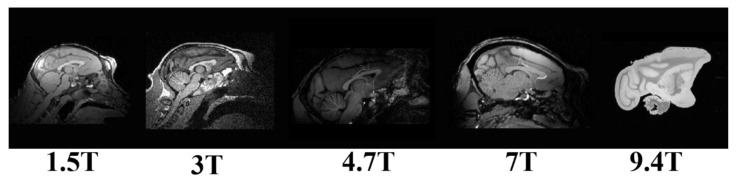
Schematic diagram of signal difference of macaque brain on 1.5 T, 3 T, 4.7 T, 7 T, and 9.4 T.

**Figure 3 brainsci-12-00260-f003:**
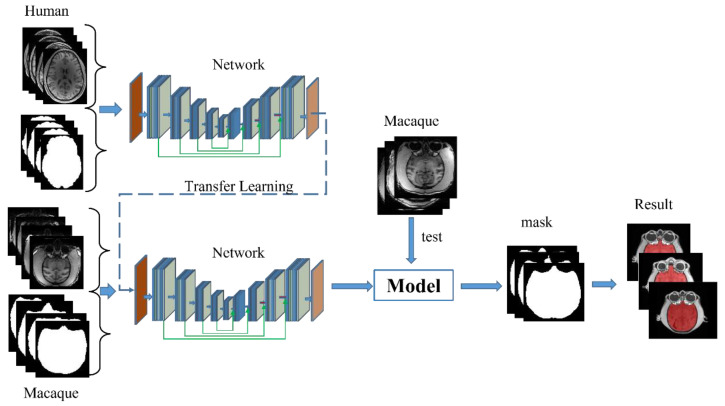
The procedure of training with the transfer learning strategy.

**Figure 4 brainsci-12-00260-f004:**
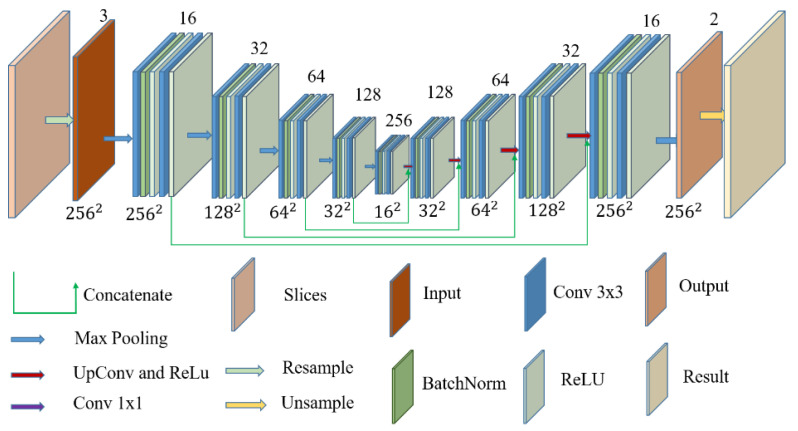
U-Net network structure.

**Figure 5 brainsci-12-00260-f005:**
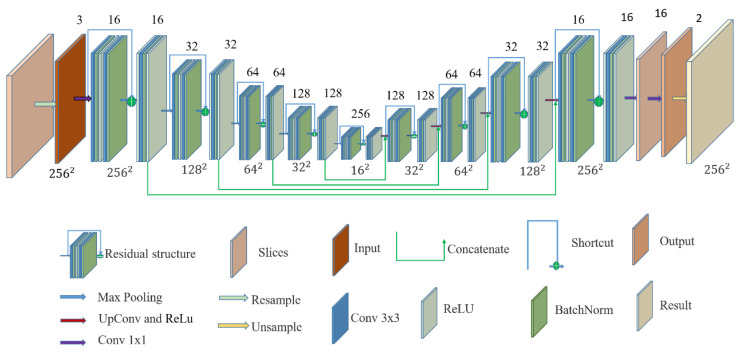
ResTLU-Net, constructed by adding a U-Net residual structure.

**Figure 6 brainsci-12-00260-f006:**
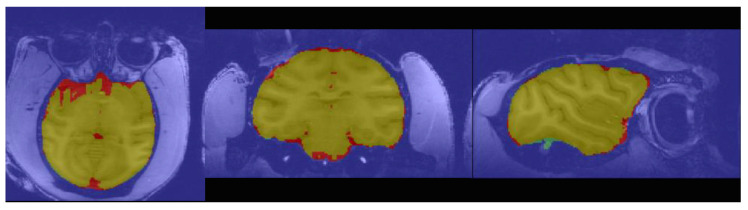
Schematic diagram of the segmentation results of the algorithm. The yellow area means true positive (*TP*); blue area means false positive (*FP*); green area means true negative (*TN*); and red area means false negative (*FN*).

**Figure 7 brainsci-12-00260-f007:**
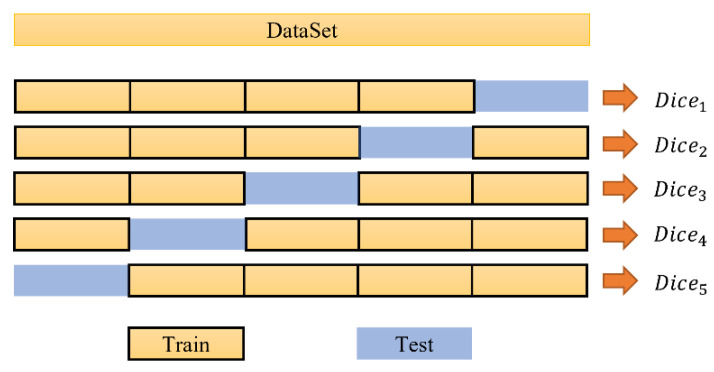
Data splitting process of five-fold cross-validation.

**Figure 8 brainsci-12-00260-f008:**
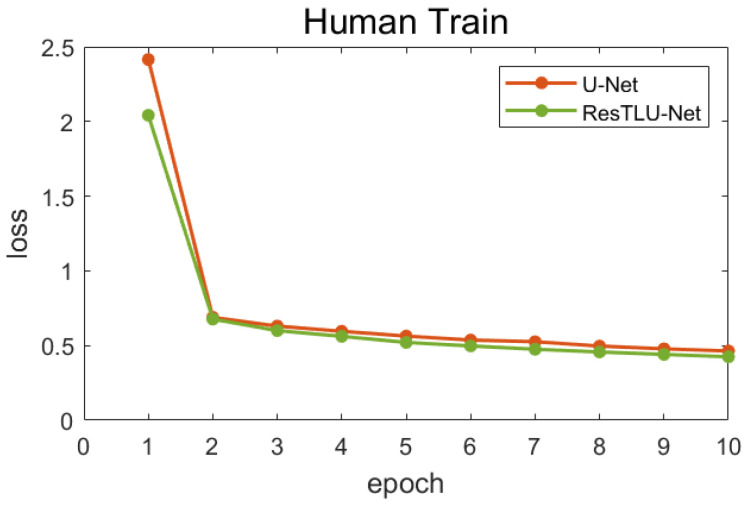
Line chart of training loss on human brain datasets (Dataset I).

**Figure 9 brainsci-12-00260-f009:**
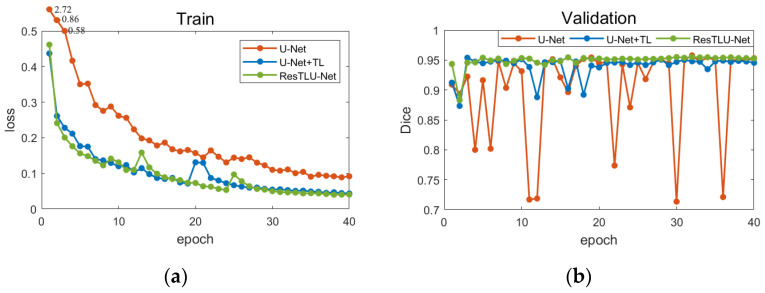
Line chart of loss value and validation result in network training. (**a**) Loss in the training process of different networks; (**b**) Dice coefficients on the validation set.

**Figure 10 brainsci-12-00260-f010:**
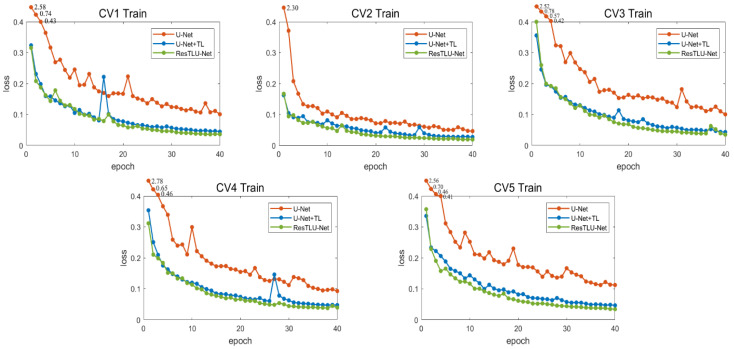
Line charts of loss value change of five-fold cross-validation comparison. The red line represents the change of loss value of U-Net; the blue line represents the change of loss value of U-Net combined with transfer learning; the green line represents the change of loss value of this ResTLU-Net. Each line chart represents the change of loss value during a cross-validation.

**Figure 11 brainsci-12-00260-f011:**
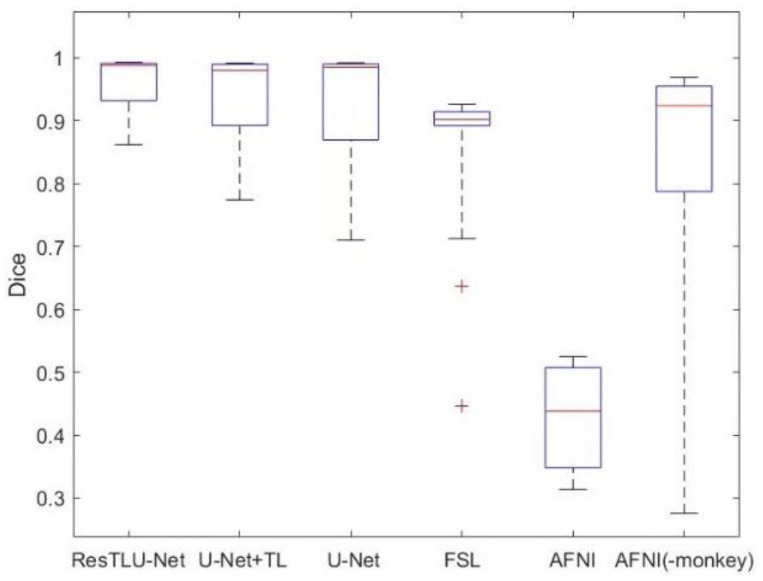
Performance of the ResTLU-Net models and other methods. The box plot shows the Dice coefficients in the testing datasets of Dataset II, across different brain extraction methods. “+” represents some particular discrete points far from the test result distribution.

**Figure 12 brainsci-12-00260-f012:**
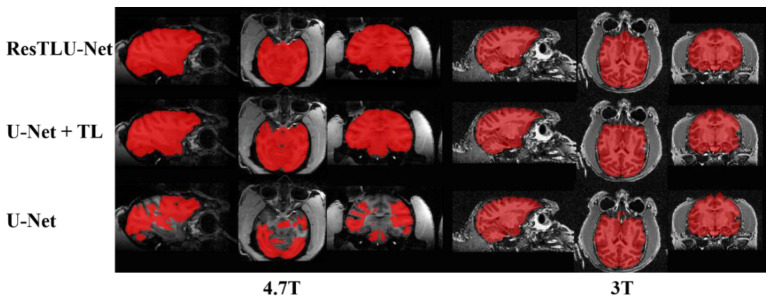
Examples of brain extraction results, using different brain extraction models.

**Table 1 brainsci-12-00260-t001:** Evaluation results of five different models.

Model	*Dice*	*TPR*	*TNR*	*Jaccard*	*VOE*	*FNR*	*FPR*	*PPV*
DenseNet	0.8372	0.7812	0.9996	0.7737	0.2263	0.2184	0.0079	0.9914
LinkNet	0.8025	0.7415	0.9998	0.7383	0.2617	0.2583	0.0035	0.9962
U-Net	0.9283	0.8826	0.9998	0.8785	0.1215	0.1172	0.0043	0.9955
U-Net + TL	0.9413	0.8991	0.9998	0.8957	0.1043	0.1006	0.0037	0.9960
ResTLU-Net	0.9581	0.9281	0.9996	0.9232	0.0768	0.0716	0.0052	0.9945

(+ TL) means model, using the transfer learning strategy.

**Table 2 brainsci-12-00260-t002:** Dice coefficient of the five-fold cross-validation results.

Model	CV1	CV2	CV3	CV4	CV5
U-Net	0.9559 ± 0.0158	0.6915 ± 0.0949	0.9427 ± 0.0097	0.9439 ± 0.0032	0.9580 ± 0.0167
U-Net + TL	0.9485 ± 0.0270	0.8409 ± 0.0741	0.9576 ± 0.0112	0.9419 ± 0.0175	0.9558 ± 0.0052
ResTLU-Net	0.9548 ± 0.0020	0.8742 ± 0.0670	0.9633 ± 0.0146	0.9410 ± 0.0031	0.9667 ± 0.0024

(+ TL) means model, using transfer learning strategy.

## Data Availability

In this research, all datasets were collected from publicly available datasets. The Human data is accessed at https://www.humanconnectome.org/study/hcp-youngadult/, accessed on 20 January 2022. The Macaque data is accessed at http://fcon_1000.projects.nitrc.org/indi/PRIMEdownloads.html, accessed on 20 January 2022.

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
