# Peer review of "A Macaque Brain Extraction Model Based on U-Net Combined with Residual Structure"

_brainsci, 2022, doi:10.3390/brainsci12020260_

Round 1

Reviewer 1 Report

in file

Reviewer 2 Report

Dear Authors,

I found your work interesting and worth publishing. Nevertheless, some revision is needed.

1. Make the last paragraph of the introduction section more descriptive. Avoid a bullet list. Do not write about the "novelty", but about what your study showed and why it is important.

2. In the 2.1. section, write that there were two datasets and then describe each. Do not start a paragraph with "Dataset I:". In such a way, it is not a sentence.

3. Do not write about your work as an "article". E.g., "This article will apply the training strategy to all models that need to be used for transfer learning." Write about your research, analysis, project, etc.

4. In Figs. 6 and 7, the data is out of scale. Find some solution to eliminate this problem, or even move the title above the data.

5. Fig. 6 is untitled. There are only panel titles.

6. Fig. 7 has no panel markings or captions for each of them.

7. The discussion is too short and too "shallow". You need to deepen it. For example, describe the limitations of your study, indicate specific directions of future research, discuss the application of the proprietary approach and the limitations of these applications in more depth. I propose to divide the discussion into sections showing the theoretical contexts, limitations and future research, and practical implications.

8. Definitely shorten the conclusions so that they are one short paragraph.

9. Review the manuscript linguistically. There are many errors, e.g., "Finally, Our model overcomes".

Sincerely,

Reviewer 3 Report

The manuscript submitted by Wang and co authors is interesting as people may be familiar with human brains as observed with MRI and other modalities but not so much with other primate brains such as the macaque. Equally important is that the majority of technical approaches for segmentation, classification, etc will be biased towards human brains and as such, this publication is of interest for those who work with non-human subjects.

There are several issues that I think will strengthen the manuscript:

  • English language, although reasonably good, should be improved. For instance, In the abstract L16 “ …modes were combined …” L17 “Then we add…” if the authors write in third person past tense (were combined) they should be consistent and write “Then, a residual network was added …”. This will apply to the whole manuscript. Other typos include lack of space before brackets (L30, L43) .
  • As the main selling point of the manuscript is the bias towards human brains, the introduction about the macaques should be expanded. Specifically, a figure that illustrates differences between macaques and humans should become Fig 1 and the description in lines 43-45 expanded. The differences due to the magnetic fields is also important, what would be the difference on the images when using T1/T2 1.5T, 4.7T, 7T, etc, this could be figure 2. Why use T2 for the training when the data of the macaques is only T1?
  • Perhaps even illustrating (no need to analyse, just illustrate in the introduction) a third species could be relevant, e.g., chimps as there may be researchers working with other primates that can see this as a potential tool even if they do not work with macaques.
  • The introduction is confusing as first argues against CNNs in favour of FCN, but then uses U-Net, which is not a FCN. This should be corrected.
  • L82-83 “…involved in the training dataset (????=0.923±0.025).” These results are completely out of context and unnecessary.
  • In section 2.5 A figure illustrating the TP,TN,FP,FN as detected in a macaque brain should be presented with the corresponding metrics.
  • Figure 9 is perhaps the most important figure and it is just so small that it is very hard to appreciate. BET and AFNI are not that important as these were not the main focus of the paper, actually BET is not presented in the introduction and HWA is. Authors could keep this one as a general results but then present another figure with fewer cases so that the actual results are much larger and the differenves between U-net, U-net TL and ResTLU can be appreciated.

Reviewer 4 Report

Reading the manuscript written by Wang et al., was really interesting. The work focuses on better using the structural similarity between humans and macaque and training deep learning networks under multicenter and limited sample data to obtain models with better generalization performance.

This article is written in a concise and orderly manner. The methodologies are appropriate and aligned with the proposed objectives. The massage from this manuscript is quite meaningful. However, I have several comments:

  • typographical errors should be corrected;
  • please provide the full name of the abbreviation at first use;
  • This new transfer learning training strategy can avoid parameter-selection bias and streamline pre-processing steps when analyzing high resolution?

Round 2

Reviewer 1 Report

Dear Authors,

The answers are partial, I understand that an attempt to include in the discussion the possible impact of systematic errors on the quality of work (AI networks) is quite difficult and requires a deeper analysis. Nevertheless, I suggest giving an example of the BSD-DTI type (or other well-defined such as "eddy current" errors) as the problem class to be considered in future research. First, we should get to the real "ground-truth" with the data, and then teach AI networks.
Good luck